# Iterative Least Trimmed Squares for Mixed Linear Regression

**Yanyao Shen**
ECE Department
University of Texas at Austin
Austin, TX 78712
shenyanyao@utexas.edu

**Sujay Sanghavi**
ECE Department
University of Texas at Austin
Austin, TX 78712
sanghavi@mail.utexas.edu

## Abstract

Given a linear regression setting, Iterative Least Trimmed Squares (ILTS) involves alternating between (a) selecting the subset of samples with lowest current loss, and (b) re-fitting the linear model only on that subset. Both steps are very fast and simple. In this paper we analyze ILTS in the setting of mixed linear regression with corruptions (MLR-C). We first establish deterministic conditions (on the features etc.) under which the ILTS iterate converges linearly to the closest mixture component. We also evaluate it for the widely studied setting of isotropic Gaussian features, and establish that we match or better existing results in terms of sample complexity. We then provide a global algorithm that uses ILTS as a subroutine, to fully solve mixed linear regressions with corruptions. Finally, we provide an ODE analysis for a gradient-descent variant of ILTS that has optimal time complexity. Our results provide initial theoretical evidence that iteratively fitting to the best subset of samples – a potentially widely applicable idea – can provably provide state-of-the-art performance in bad training data settings.

## 1 Introduction

In vanilla linear regression, one (implicitly) assumes that each sample is a linear measurement of a single unknown vector, which needs to be recovered from these measurements. Statistically, it is typically studied in the setting where the samples come from such a ground truth unknown vector, and we are interested in the (computational/statistical complexity of) recovery of this ground truth vector. **Mixed linear regression** (MLR for brevity) is the problem where there are *multiple* unknown vectors, and each sample can come from any one of them (and we do not know which one, a-priori). Our objective is again to recover all (or some, or one) of them from the samples. In this paper, we consider MLR with the additional presence of *corruptions* – i.e. adversarial additive errors in the responses – for some unknown subset of the samples. There is now a healthy and quickly growing body of work on algorithms, and corresponding theoretical guarantees, for MLR with and without additive noise and corruptions; we review these in detail in the related work section.

In our paper we start from a classical (but hard to compute) approach from robust statistics: **least trimmed squares** [19]. This advocates fitting a model so as to minimize the loss on only a fraction $\tau$ of the samples, instead of all of them – but crucially, the subset $S$ of samples chosen and the model to fit them are to be estimated *jointly*. To be more specific, suppose our samples are $(x_i, y_i)$, for $i \in [n]$.

Then the least squares (LS) and least trimmed squares (LTS) estimates are:

$$\widehat{\theta}_{\text{LS}} = \arg\min_\theta \sum_{i\in[n]} \left(y_i - \langle x_i, \theta\rangle\right)^2,$$

$$\widehat{\theta}_{\text{LTS}} = \arg\min_\theta \min_{S\,:\,|S|=\lfloor\tau n\rfloor} \sum_{i\in S} \left(y_i - \langle x_i, \theta\rangle\right)^2.$$

Note that least trimmed squares involves a parameter: the fraction $\tau$ of samples we want to fit. Solving for the least trimmed squares estimate $\widehat{\theta}_{\text{LTS}}$ needs to address the combinatorial issue of finding the best subset to fit, but the goodness of a subset is only known once it is fit. LTS is shown to have computation lower bound exponential in the dimension of $x$ [17].

LTS, if one could solve it, would be a candidate algorithm for MLR as follows: suppose we knew a lower bound on the number of samples corresponding to a single component (i.e. generated using one of the unknown vectors). Then one would choose the fraction $\tau$ in the LTS procedure to be *smaller* than this lower bound on the fraction of samples that belong to a component. Ideally, this would lead the LTS to choose a subset $S$ of samples that all correspond to a single component, and the least squares on that set $S$ would find the corresponding unknown vector. This is easiest to see in the noiseless corruption-less setting where each sample is just a pure linear equation in the corresponding unknown vector. In this case, an $S$ containing samples only from one component, and a $\theta$ which is the corresponding ground truth vector, would give 0 error and hence would be the best solutions to LTS. Hence, to summarize, one can use LTS to solve MLR by estimating **a single ground truth vector at a time**.

However, LTS is intractable, and we instead study the natural iterative variant of LTS, which alternates between finding the set $S \subset n$ of samples to be fit, and the $\theta$ that fits it. In particular, our procedure – which we call **iterative least trimmed squares (ILTS)** – first picks a fraction $\tau$ and then proceeds in iterations (denoted by $t$) as follows: starting from an initial $\theta_0$,

$$S_t = \arg\min_{S\,:\,|S|=\lfloor\tau n\rfloor} \sum_{i\in S} \left(y_i - \langle x_i, \theta_t\rangle\right)^2,$$

$$\theta_{t+1} = \arg\min_\theta \sum_{i\in S_t} \left(y_i - \langle x_i, \theta\rangle\right)^2.$$

Note that now, as opposed to before, finding the subset $S_t$ is trivial: just sort the samples by their current squared errors $(y_i - \langle x_i, \theta_t\rangle)^2$, and pick the $\tau n$ that have smallest loss. Similarly, the $\theta$ update now is a simple least squared problem on a pre-selected subset of samples. Note also that each of the above steps decreases the function $a(\theta, S) \triangleq \sum_{i\in S} \left(y_i - \langle x_i, \theta\rangle\right)^2$. This has also been referred to as iterative hard thresholding and studied for the different but related problem of robust regression, again please see related work for known results. Our motivations for studying ILTS are several: *(1)* it is very simple and natural, and easy to implement in much more general scenarios beyond least squares. Linear regression represents in some sense the simplest statistical setting to understand this approach. *(2)* In spite of its simplicity, we show in the following that it manages to get state-of-the-art performance for MLR with corruptions, with weaker assumptions than several existing results.

Again as before, one can use ILTS for MLR by choosing a $\tau$ that is smaller than the number of samples in a component. However, additionally, we now also need to choose an initial $\theta_0$ that is closer to one component than the others. In the following, we thus give two kinds of theoretical guarantees on its performance: a local one that shows linear convergence to the closest ground truth vector, and a global one that adds a step for good initialization.

**Main contributions and outline:**

- We propose a simple and efficient algorithm ILTS for solving MLR with adversarial corruptions; we precisely describe the problem setting in **Section 3**. ILTS starts with an initial estimate of a single unknown $\theta$, and alternates between selecting the size $\tau n$ subset of the samples best explained by $\theta$, and updating the $\theta$ to best fit this set. Each of these steps is very fast and easy.

- Our first result, **Theorem 4** in **Section 4** establishes deterministic conditions – on the features, the initialization, and the numbers of samples in each component – under which ILTS linearly converges to the ground truth vector that is closest to the initialization. **Theorem 7** in **Section 4** specializes this to the (widely studied) case when the features are isotropic Gaussians. The

sample complexity is nearly optimal in both dimension $d$ and the number of components $m$, while previous state-of-the-art results are nearly optimal in $d$, but can be exponential in $m$. Our analysis for inputs following isotropic Gaussian distribution is easy to generalize to more general class of sub-Gaussian distributions.

- To solve the full MLR problem, we identify *finding the subspace spanned by the true MLR components* as a core problem for initialization. In the case of isotropic Gaussian features, this is known to be possible by existing results in robust PCA (when corruptions exist) or standard spectral methods (when there are no corruptions). Given a good approximation of this subspace, one can use the ILTS process above as a subroutine with an "outer loop" that tries out many initializations (which can be done in parallel, and are not too many when number of components is fixed and small) and evaluates whether the final estimate is to be accepted as an estimate for a ground truth vector (Global-ILTS). We specify and analyze it in **Section 5** for the case of random isotropic Gaussian features and also discuss the feasibility of finding such a subspace.

## 2 Related Work

**Mixed linear regression** Learning MLR even in the two mixture setting is NP hard in general [26]. As a result, provably efficient algorithmic solutions under natural assumptions on the data, e.g., all inputs are i.i.d. isotropic Gaussian, are studied. Efficient algorithms that provably find both components include the idea of using spectral initialization with local alternating minimization [26], and classical EM approach with finer analysis [1, 12, 13]. In the multiple components setting, substituting spectral initialization by tensor decomposition brings provable algorithmic solutions [5, 27, 29, 20]. Recently, [14] proposes an algorithm with nearly optimal complexity using quite different ideas. They relate MLR problem with learning GMMs and use the black-box algorithm in [16]. In Table 1, we summarize the sample and computation complexity of the three most related work. Previous literature focus on the dependency on dimension $d$, for all these algorithms that achieve near optimal sample complexity, the dependencies on $m$ for all the algorithms are expoential (notice that the guarantees in [27] contains a $\sigma_m$ term, which can be exponentially small in $m$ without further assumptions, as pointed out by [14]), and [14] requires exponential in $m^2$ number of samples for a more general class of Gaussian distributions. Notice that while it is reasonable to assume $m$ being a constant, this exponential dependency on $m$ or $m^2$ could dominate the sample complexity in practice. From robustness point of view, the analysis of all these algorithms rely heavily on exact model assumptions and are restricted to Gaussian distributions. While recent approaches on robust algorithms are able to deal with strongly convex functions, e.g., [9], with corruption in both inputs and outputs, [29] showed local strong convexity of MLR with small local region $\tilde{\mathcal{O}}(d(md)^{-m})$, under $\tilde{\Omega}(dm^m)$ samples. To the best of our knowledge, we are not aware of any previous work study the algorithmic behavior under mis-specified MLR model settings. We provide fine-grained analysis for a simple algorithm that achieves nearly optimal sample and computation complexity.

**Robust regression** Our algorithm idea is similar to least trimmed square estimator (LTS) proposed by [19]. The hardness of finding the exact LTS estimator is discussed in [17], which shows an exponential in $d$ computation lower bound under the hardness of affine degeneracy conjecture. While our algorithm is similar to the previous hard thresholding solutions proposed in [2], their analysis does not handle the MLR setting, and only guarantees parameter recovery given a small constant fraction of corruption. Algorithmic solutions based on LTS for solving more general problems have been proposed in [25, 23, 21]. [10] studies the $l_1$ regression and gives a tight analysis for recoverable corruption ratio. Another line of research focus on robust regression where both the inputs and outputs can be corrupted, e.g., [6]. There are provable recovery guarantees under constant ratio of corruption using using robust gradient methods [9, 18, 15], and sum of squares method [11]. We focus on computationally efficient method with nearly optimal computation time that is easily scalable in practice.

## 3 Problem Setup and Preliminaries

We consider the standard (noiseless) MLR model with corruptions, which we will abbreviate to **(MLR-C)**; each sample is a linear measurement of one of $m$ unknown "ground truth" vectors – but we do not know which one. Our task is to find the ground truth vectors, and this is made harder by a constant fraction of all samples having an additional error in responses. We now specify this formally.

| | setting | | sample $(n)$ | computation |
|---|---|---|---|---|
| [27] | $\mathcal{N}(0,\mathbf{I}_d)$, $\sigma_k$ <br> linearly independent $\theta_{(j)}$s | local <br> global | $\texttt{poly}(m)d$ <br> $\texttt{poly}(m)d/\sigma_m^5$ | $nd^2+md^3$ <br> $nd^2+\texttt{poly}(m)$ |
| [29] | $\mathcal{N}(0,\mathbf{I}_d)$, constant $Q$ | | $m^m d$ | $nd$ |
| [14] | $\mathcal{N}(0,\Sigma_{(j)})$ | | $d\texttt{poly}(\frac{m}{Q}) + (\frac{cm}{Q})^{m^2}$ | $nd$ |
| Ours | robust, not limited to $\mathcal{N}(0,\Sigma_{(j)})$ <br> good estimate of the subspace | local <br> global | $md$ <br> - | $nd^2$ ($nd$ for GD-ILTS) <br> subspace est. $+(\frac{cm}{Q})^m \cdot nd$ |

Table 1: Compare with previous results in the setting of balanced MLR, i.e., each component has $n/m$ samples. $Q$ represents a separation property of the mixture components (see Definition 1 for details). For conciseness, we only keep the main factors in the complexity terms. The inspiring algorithms listed here achieve nearly optimal sample complexity (nearly linear in $d$) under certain settings, which are helpful for understanding the limit of learning MLR. Note that we have $\tilde{\mathcal{O}}(nd^2 \log \frac{1}{\varepsilon})$ computation for ILTS and $\tilde{\mathcal{O}}(nd \log^2 \frac{1}{\varepsilon})$ for GD-ILTS (in Section B, a direct gradient variant). Sample complexity for our global step depends on the hardness of finding the subspace. Our local requirement only needs the current estimation to be close to one of the components, which is much easier to satisfy than the local notion in [27]. Methods in [5, 20] require $\tilde{\Omega}(d^6)$ and $\tilde{\Omega}(d^3)$ sample complexity (they can handle more general settings), [28] uses sparse graph codes for sparse MLR. Therefore, we do not list their results here (hard to compare with).

**(MLR-C)**: We are given $n$ samples of the form $(x_i, y_i)$ for $i = 1, \ldots, n$, where each $y_i \in \mathbb{R}$ and $x_i \in \mathbb{R}^d$. Unknown to us, there are $m$ "ground truth" vectors $\theta^\star_{(1)}, \ldots, \theta^\star_{(m)}$, each in $\mathbb{R}^d$; correspondingly, and again unknown to us, the set of samples is partitioned into disjoint sets $S_{(1)}, \ldots, S_{(m)}$. If the $i^{th}$ sample is in set $S_{(j)}$ for some $j \in [m]$, it satisfies

$$y_i = \langle x_i, \theta^\star_{(j)} \rangle + r_i, \quad \text{for } i \in S_{(j)} \quad \textbf{(MLR-C)}.$$

Here, $r_i$ denotes the possible additive corruption – a fraction of the $r_1, \ldots, r_n$ are arbitrary unknown values, and the remaining are 0 (and again, we are not told which).

Our **objective** is: given only the samples $(x_i, y_i)$, find the ground truth vectors $\theta^\star_{(1)}, \ldots, \theta^\star_{(m)}$. In particular, we do not have a-priori knowledge of any of the sets $S_{(j)}$, or the values/support of the corruptions. We now develop some notation for the sizes of the components etc.

**Sizes of sets:** Let $R^\star = \{i \in [n] \text{ s.t. } r_i \neq 0\}$ denote the set of corrupted samples; note that this set can overlap with any / all of the components' sets $S_{(j)}$s. Let $S^\star_{(j)} = S_{(j)} \setminus R^\star$ be the uncorrupted set of samples from the $S_{(j)}$, for all $j \in [m]$. Let $\tau^\star_{(j)} = |S^\star_{(j)}|/n$ denote the fraction of uncorrupted samples in each component $j$, and $\tau^\star_{\min} = \min_{j \in [m]} \tau^\star_{(j)}$ denote the smallest such fraction. Let $\gamma^\star = |R^\star|/(n\tau^\star_{\min})$ be the ratio of the number of corrupted samples to the size of the smallest component[1]. Notice that $\gamma^\star = 0$ corresponds to the MLR model without corruption. We do not make any assumptions on which specific samples are corrupted; $R^\star$ can be any subset of size $\gamma^\star \tau^\star_{\min} n$ of the set of $n$ samples. Thus a $\gamma^\star = 1$ situation can prevent the recovery of the smallest component.

Finally, for convenience, we denote $S^\star_{(-j)} := \cup_{l \in [m] \setminus \{j\}} S^\star_{(l)}$, $\forall j \in [m]$, $\mathbf{X} = [x_1, \cdots, x_n]^\top \in \mathbb{R}^{n \times d}$, and $y = [y_1, \cdots y_n]$. Note that we consider the case without additive stochastic noise, which is the same setting as in [27, 29, 14].

## 3.1 Preliminaries

We now develop our way to making a few basic assumptions on the model setting; our main results show that under these assumptions the simple ILTS algorithm succeeds. The first definition quantifies the separation between the ground truth vectors.

**Algorithm 1** ILTS (for recovering a single component)
---
1: **Input:** Samples $\mathcal{D}_n = \{x_i, y_i\}_{i=1}^n$, initial $\theta_0$, fraction of samples to be retained $\tau$
2: **Output:** Final estimation $\widehat{\theta}$
3: **Parameters:** Number of rounds $T$
4: **for** $t = 0$ to $T - 1$ **do**
5:     $S_t \leftarrow$ index set of $\lfloor \tau n \rfloor$ samples with smallest residuals $(y_i - \langle x_i, \theta_t \rangle)^2$, $i \in [n]$
6:     $\theta_{t+1} = \arg \min_\theta \sum_{i \in S_t} (y_i - \langle x_i, \theta \rangle)^2$
7: **Output:** $\widehat{\theta} = \theta_T$
---

**Definition 1** (Q-separation). *For the set of components $\{\theta_{(1)}^\star, \cdots, \theta_{(m)}^\star\}$,*

*(i) the set of components is Q-separated if $Q \leq \frac{\min_{i,j \in [m], i \neq j} \|\theta_{(i)}^\star - \theta_{(j)}^\star\|_2}{\max_{j \in [m]} \|\theta_{(j)}^\star\|}$;*

*(ii) local separation $Q_j$ is defined as $Q_j = \frac{\min_{l \in [m] \setminus \{j\}} \|\theta_{(l)}^\star - \theta_{(j)}^\star\|_2}{\|\theta_{(j)}^\star\|}, \forall j \in [m]$.*

By definition, it is clear that $Q \leq Q_j, \forall j \in [m]$. In fact, $Q$ represents the global separation property, which is required by previous literatures for solving MLR [27, 29, 14], while $Q_j$ describes the local separation property for the $j^{th}$ component, and gives us a better characterization of the local convergence property for a single component. We now turn to the features; let $\mathbf{X}$ denote the $n \times d$ matrix of features, with the $i^{th}$ row being $x_i^\top$ – the features of the $i^{th}$ sample.

**Definition 2** (($\psi^+, \psi^-$)-feature regularity). *Define $\mathcal{S}_k$ to be the set of all subsets in $[n]$ with size $k$, and let $\mathbf{X}_S$ be the sub-matrix of $\mathbf{X}$ with rows indexed by some $S \subset [n]$. Define*

$$\psi^+(k) = \max_{S \in \mathcal{S}_k} \sigma_{\max}(\mathbf{X}_S^\top \mathbf{X}_S), \quad and \quad \psi^-(k) = \min_{S \in \mathcal{S}_k} \sigma_{\min}(\mathbf{X}_S^\top \mathbf{X}_S), \tag{1}$$

*where functions $\psi^+(k), \psi^-(k)$ are feature regularity upper bound and lower bound, respectively. $\sigma_{\max}(\mathbf{A})(\sigma_{\min}(\mathbf{A}))$ represents the largest (smallest) eigenvalue of a symmetric matrix $\mathbf{A}$.*

Clearly, if $\psi^+$ is too large or $\psi^-$ is too small, identifying samples belonging to a certain component or not, even given a very good estimate of the true component, can become extremely difficult. For example, if the true component coincides with the top eigenvalue direction of its feature covariance matrix, then, even if the current estimate is close within $\ell_2$, the prediction error can still be quite large due to the $\mathbf{X}$. On the other hand, if each row in $\mathbf{X}$ follows i.i.d. isotropic Gaussian distribution, $\psi^+(k)$ and $\psi^-(k)$ are upper and lower bounded by $\Theta(n)$ for $k$ being a constant factor of $n$ (when $n$ is large enough). This is shown in Lemma 5. Next, we define $\Delta$-affine error, a property of the data that is closely connected with our analysis of ILTS in Section 4.

**Definition 3** ($\Delta$-affine error $/\mathcal{V}(\Delta)$). *For $\forall j \in [m]$, denote $\mathbf{X}_{(j)}$ as the sub-matrix with rows from $S_{(j)}^\star$ with size $n\tau_{(j)}^\star$, $\mathbf{X}_{(-j)}$ as the sub-matrix with rows from $S_{(-j)}^\star$ with size $\tau_{(-j)}^\star$, let $\tau_{(j)} = c_\tau \tau_{(j)}^\star$ for some fixed constant $c_\tau < 1$. Define $\Delta$-affine error $\mathcal{V}(\Delta)$ to be the maximum value of integer $V$ such that the following holds for some $v_1, v_2 \in \mathbb{R}^d$ with $\|v_1\|_2/\|v_2\|_2 = \Delta \leq 1$ and $j \in [m]$:*

$$[|\mathbf{X}_{(j)} v_1|]_{(V + \lceil (\tau_j^\star - \tau_j) n \rceil)^{\text{th}} \ largest} \geq [|\mathbf{X}_{(-j)} v_2|]_{(V)^{\text{th}} \ smallest}. \tag{2}$$

This is saying, when we pick samples from set $S_{(j)}^\star$ and $S_{(-j)}^\star$ by ranking and finding the smallest $\lfloor \tau_j n \rfloor$ samples based on the projected values to $v_1, v_2$, the number of samples from $S_{(-j)}^\star$ is at most $\mathcal{V}(\Delta)$. For example, given current estimate $\theta$, the residual of a sample from component $j$ is $\langle x_i, \theta_{(j)}^\star - \theta \rangle$, and $v_1$ can be considered as $\theta_{(j)}^\star - \theta$. As a result, this definition helps quantify the number of mis-classified samples from other components, see Figure 1 for another illustration. If each row in $\mathbf{X}$ follows i.i.d. isotropic Gaussian distribution, $\mathcal{V}(\Delta)$ scales linearly with $\Delta$ for large enough $n$. This is shown in Lemma 6.

## 4   ILTS and Local Analysis

Algorithm 1 presents the procedure of ILTS: Starting from initial parameter $\theta_0$, the algorithm alternates between (a) selecting samples with smallest residuals, and (b) getting the least square

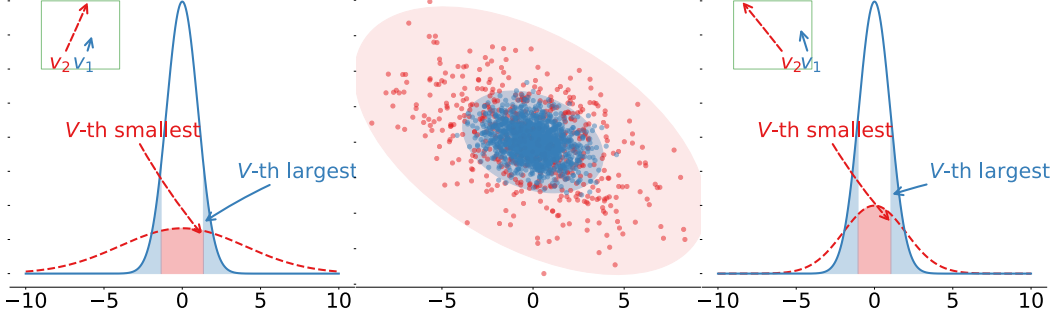

Figure 1: A two-dimensional illustration of $\Delta$-affine error $\mathcal{V}(\Delta)$ in Definition 3 for $\|v_1\|_2/\|v_2\|_2 = \Delta$ (for simplicity, assume $\tau_j^\star = \tau_j$). $\mathcal{V}(\Delta)$ can be interpreted as the number of mistakenly filtered samples in **any** directions. The plot in the middle contains blue dots from one component $\mathbf{X}_{(j)}$, and red dots from other components $\mathbf{X}_{(-j)}$. The plots on the left and right illustrate how the histogram looks like for $\mathbf{X}_{(j)}v_1$ (in blue) and $\mathbf{X}_{(-j)}v_2$ (in red), for two sets of $v_1$ and $v_2$. Areas in blue represent the samples that may be mistakenly filtered out. The maximum $V$ that satisfies (2) is larger on the right side plot since projected values for samples from $S_{(-j)}^\star$ are more concentrated. $\mathcal{V}(\Delta)$ is an upper bound of the maximum $V$ on all possible directions.

solution on the selected set of samples as the new parameter. Intuitively, ILTS succeeds if (a) $\theta_0$ is close to the targeted component, and (b) for each round of update, the new parameter is getting closer to the targeted component. For our analysis, we assume the chosen fraction of samples to be retained is strictly less than the number of samples from the interested component, i.e., $\tau = c_0\tau_{(j)}^\star$ for some universal constant $c_0$. We first provide local recovery results using the structural definition we made in Section 3, for both no corruption setting and corruption setting. Then, we present the result under Gaussian design matrix. All proofs can be found in the Appendix.

**Theorem 4** (deterministic features). *Consider (**MLR-C**) using Algorithm 1 with $\tau < \tau_{(j)}^\star$. Given iterate $\theta_t$ at round $t$, which is closer to the $j$-th component in Euclidean distance and satisfies $\|\theta_t - \theta_{(j)}^\star\|_2 \leq \frac{1}{2}\min_{l \in [m]\setminus\{j\}}\|\theta_{(j)}^\star - \theta_{(l)}^\star\|_2$, then the next iterate $\theta_{t+1}$ of the algorithm satisfies*

$$\|\theta_{t+1} - \theta_{(j)}^\star\|_2 \leq \frac{2\psi^+\left(\mathcal{V}\left(\frac{1}{Q_j} \cdot \frac{2\|\theta_t - \theta_{(j)}^\star\|_2}{\|\theta_{(j)}^\star\|_2}\right) + \gamma^\star\tau_{\min}^\star n\right)}{\psi^-(\tau n)}\|\theta_t - \theta_{(j)}^\star\|_2. \tag{3}$$

The above one-step update rule (3) holds as long as Algorithm 1 uses $\tau < \tau_{(j)}^\star$ and the iterate $\theta_t$ is closer to the $j$-th component. However, in order to make $\theta_{t+1}$ getting closer to $\theta_{(j)}^\star$, the contraction term on the RHS of (3) needs to be less than 1, which may require stronger conditions on $\theta_t$, depending on what $x_i$s are. The denominator term $\psi^-(\tau n)$ is due to the selection bias on a subset of samples, which scales with $n$ as long as the inputs have good regularity property. The numerator term is due to the incorrect samples selected by $S_t$, which consists of: (a) samples from other mixture components, and (b) corrupted samples. (a) is controlled by the affine error, which depends on (a1) the local separation of components $Q_j$, and (a2) the relative closeness of $\theta_t$ to $\theta_{(j)}^\star$, and scales with $n$. The affine error $\mathcal{V}$ gets larger if the separation is small, or $\theta_t$ is not close enough to $\theta_{(j)}^\star$. For (b), the number of all corrupted samples is controlled by $\gamma^\star\tau_{\min}^\star n$, which is not large given $\gamma^\star$ being a small constant.

Theorem 4 gives a general update rule for any given dataset according to Definitions 1–2. Next, we present the local convergence result for the specific setting of Gaussian input vectors, by giving a tight analysis for feature regularity in Lemma 5 and a tight bound for the affine error $\mathcal{V}(\Delta)$ in Lemma 6.

**Lemma 5.** *Let $\psi^+(k), \psi^-(k)$ be defined as in (1), and assume each $x_i \sim \mathcal{N}(0, \mathbf{I}_d)$. Then, for $k = c_k n$ with constant $c_k$, for $n = \Omega\left(\frac{d\log\frac{1}{c_k}}{c_k}\right)$, with high probability,*

$$\psi^+(k) \leq c_1 \cdot k, \quad \psi^-(k) \geq c_2 \cdot k,$$

where $c_1, c_2$ are constants that depend on $c_k$: $c_1 \leq 1 + 3e\sqrt{6\log\frac{2}{c_k}} + \frac{C_1}{c_k}$, $c_2 \geq C_2 c_k$, for universal constants $C_1, C_2$.

**Lemma 6.** *Suppose we have $x_i \sim \mathcal{N}(0, \mathbf{I}_d)$, $\tau^\star_{(j)} n$ samples for each class $S_{(j)}$. Then, for $n = \Omega\left(d\log\log d/\tau^\star_{\min}\right)$, with high probability, the design matrix satisfies $\mathcal{V}(\Delta) \leq c\left\{\Delta n \vee \log n\right\}$.*

Plug in Lemma 5 and Lemma 6 to Theorem 4, we have:

**Theorem 7** (Gaussian features)**.** *For (MLR-C), assume $x_i \sim \mathcal{N}(0, \mathbf{I}_d)$, consider using Algorithm 1 with $\tau < \tau^\star_{(j)}$, $\forall j \in [m]$, and $n = \Omega\left(\frac{d\log\log d}{\tau^\star_{\min}}\right)$. If the iterate satisfies $\|\theta_t - \theta^\star_{(j)}\| \leq c_j \min_{l \in [m]\setminus\{j\}} \|\theta^\star_{(l)} - \theta^\star_{(j)}\|_2$ (where $c_j$ is a constant depending on $\tau$ and $\tau^\star_{(j)}$) for some $j \in [m]$, then, w.h.p., the next iterate $\theta_{t+1}$ of the algorithm satisfies*

$$\|\theta_{t+1} - \theta^\star_{(j)}\|_2 \leq \kappa_t \|\theta_t - \theta^\star_{(j)}\|_2, \tag{4}$$

*where $\kappa_t = \frac{c_0}{\tau n}\left(\left\{\frac{1}{Q_j} \cdot \frac{2\|\theta_t - \theta^\star_{(j)}\|_2}{\|\theta^\star_{(j)}\|_2} n \vee \log n\right\} + \gamma^\star \tau^\star_{\min} n\right) < 1$, for some small constant $\gamma^\star$.*

Note that in this Theorem, $c_0$ is a constant such that $\kappa_t < 1$, and such a $c_0$ corresponds to an upper bound on $c_j$, i.e., the local region. Theorem 7 shows that, as long as $\theta_t$ is contant time closer to $\theta^\star_{(j)}$, we can recover $\theta^\star_{(j)}$ up to arbitrary accuracy with $\tilde{\mathcal{O}}(d/\tau^\star_{\min})$ samples. In fact, Lemma 5 and Lemma 6 (and hence Theorem 7) are generalizable to more general distributions, including the setting studied in [14]. The initial condition simply changes by a factor of $\sigma$, where $\sigma$ is the upper bound of the covariance matrix. The formal statement is as follows:

**Corollary 8** (features with non-isotropic Gaussians)**.** *Consider (MLR-C), where each $x_i \sim \mathcal{N}(0, \Sigma_{(j)})$ for $i \in S_{(j)}$, $\mathbf{I} \preceq \Sigma_{(j)} \preceq \sigma\mathbf{I}$. Under the same setting as in Theorem 7, convergence property (4) holds as long as iterate $\theta_t$ satisfies $\|\theta_t - \theta^\star_{(j)}\| \leq c_j \frac{\tau}{\sigma} \min_{l \in [m]\setminus\{j\}} \|\theta^\star_{(l)} - \theta^\star_{(j)}\|_2$.*

**Discussion** We summarize our results from the following four perspectives:

- Our results can generalize to a wide class of distributions: e.g., Gaussians or a sub-class of sub-Gaussians with different covariance matrix. This is because the proof technique for showing Lemma 5 and Lemma 6 only exploits the property of (a) concentration of order statistics; (b) anti-concentration of Gaussian-type distributions.

- Super-linear convergence speed for $\gamma^\star = 0$: When $\gamma^\star = 0$, $\kappa_t \propto \|\theta_t - \theta^\star_{(j)}\|_2$ in Theorem 7.

- Optimal local sample dependency on $m$: Notice that locally, in the balanced setting, where $\tau^\star_{(j)} = 1/m$, the sample dependency on $m$ is linear. This dependency is optimal since for each component, we want $n/m > d$ to make the problem identifiable. [2]

- ILTS learns each component separately: Different from the local alternating minimization approach by [27], recovering one component does not require good estimates of any other components. E.g., if we are only interested in the $j$-th component, then, the sample complexity is $\tilde{\mathcal{O}}\left(d/\tau^\star_{(j)}\right)$.

## 5  Global ILTS  and Its Analysis

In Section 4, we show that as long as the initialization is closer to the targeted component with constant factor, we can locally recover the component, even under a constant fraction of corruptions. In this part, we discuss the initialization condition. Let us define the targeted subspace $\mathcal{U}_m$ as: $\mathcal{U}_m := \mathtt{span}\left\{\theta^\star_{(1)}, \theta^\star_{(2)}, \cdots, \theta^\star_{(m)}\right\}$, and for any subspace $\mathcal{U}$, we denote $\mathbf{U}$ as the corresponding subspace matrix, with orthonormal columns. We define the concept of $\epsilon$-close subspace as follows:

**Definition 9** ($\epsilon$-close subspace)**.** *$\widehat{\mathcal{U}} \in \mathbb{R}^{d \times \tilde{m}}$ is an $\epsilon$-close subspace to $\mathcal{U}_m$ if $\tilde{m} = \mathcal{O}(m)$, and their corresponding subspace matrices $\widehat{\mathbf{U}}, \mathbf{U}_m$ satisfy: $\left\|\left(\mathbf{I}_d - \widehat{\mathbf{U}}\widehat{\mathbf{U}}^\top\right) \cdot \mathbf{U}_m\right\|_2 \leq \epsilon$.*

**Algorithm 2** GLOBAL-ILTS (for recovering all components )

---

1: **Input**: Samples $\mathcal{D}_n = \{x_i, y_i\}_{i=1}^n$
2: **Output**: $\widehat{\theta}_1, \cdots, \widehat{\theta}_m$
3: **Parameters:** Granularity $\epsilon$, estimate $\{\tau_j\}_{j=1}^m$, small error $\delta$
4: Find a $\epsilon$-close subspace $\mathcal{U}_m$
5: Generate an $\epsilon$-net $\Theta_\epsilon$ covering the centered sphere in $\mathcal{U}_m$ with radius $\|\max_{j\in[m]} \theta_{(j)}^\star\|_2$
6: **for** $j = 1$ to $m$ **do**
7:     **for** $\tilde{\theta}$ randomly drawn from $\Theta_\epsilon$ **do**
8:        $\theta \leftarrow \text{ILTS}(\mathcal{D}_n, \tilde{\theta}, \tau_j)$
9:        $S_j = \{i \mid (y_i - \langle x_i, \theta\rangle)^2 < \delta^2\}$
10:       **if** $|S_j| \geq \lfloor \tau_j n \rfloor$ **then**
11:         $\widehat{\theta}_j = \theta$, **break**
12:     Remove samples in set $S_j$ from $\mathcal{D}_n$
13: **Return:** $\widehat{\theta}_1, \cdots, \widehat{\theta}_m$

---

An interpretation of an $\epsilon$-close subspace $\mathcal{U}$ is as follows: for any unit vector $v$ from subspace $\mathcal{U}_m$, there exists a vector $v'$ in subspace $\mathcal{U}$ with norm less than 1, such that $\|v - v'\|_2 \leq \epsilon$. We also define $\varepsilon$-recovery, to help with stating our theorem.

**Definition 10** ($\varepsilon$-recovery). $\widehat{\Theta} = \left[\widehat{\theta}_1, \cdots, \widehat{\theta}_m\right]$ *is a $\varepsilon$-recovery of* $\Theta^\star = \left[\theta_{(1)}^\star, \cdots, \theta_{(m)}^\star\right]$ *if* $\min_{\mathbf{P}\in\mathcal{P}_m} \|\widehat{\Theta}\mathbf{P} - \Theta^\star\|_{2,\infty} \leq \varepsilon$, *where $\mathcal{P}_m$ is the class of all $m$-dimensional permutation matrices.*

The procedure for Global-ILTS is shown in Algorithm 2. The algorithm takes a subspace as its input, which should be a good approximation of the subspace spanned by the correct $\theta_{(j)}^\star$s. Given the subspace, Global-ILTS constructs an $\epsilon$-net over a sphere in subspace $\mathcal{U}_m$ The algorithm then iteratively removes samples once the ILTS sub-routine finds a valid component. Notice that we require the estimates $\tau_j$s to satisfy $\tau_j < \tau_{(j)}^\star$. [3] We have the following global recovery result:

**Theorem 11** (Global algorithm). *For (MLR-C), assume $x_i \sim \mathcal{N}(0, \mathbf{I}_d)$. Following Algorithm 2, we can find an $\epsilon$-close subspace $\mathcal{U}$ with $\epsilon = \frac{c_l \min_{j\in[m]} \tau_j}{2}$, and with small $\delta, \varepsilon$ (e.g., $\delta = c\sqrt{\log n}\varepsilon$ with $\varepsilon$ small enough) and $\tau_j < \tau_{(j)}^\star$ for all $j \in [m]$, we are able to have $\varepsilon$-recovery over all components with $n = \Omega\left(\frac{d\log\log d}{\tau_{\min}^\star}\right)$ samples, in $\mathcal{O}\left(\left(\frac{1}{\tau_{\min}^\star Q}\right)^{\mathcal{O}(m)} nd^2 \log\frac{1}{\varepsilon}\right)$ time.*

**Several merits of Theorem 11:** First, our result clearly separates the problem into (a) globally finding a subspace; and (b) locally recovering a single component with ILTS. Second, the $nd^2$ computation dependency is due to finding the exact least squares. Alternatively, one can take gradient descent to find an approximation to the true component. The convergence property of a gradient descent variant of ILTS is shown in Section B, where we further discuss the ideal number of gradient updates to make for each round, so that the algorithm can be more efficient. Third, the $\exp(\mathcal{O}(m))$ dependency in runtime can be practically avoided, since our algorithm is easy to run in parallel.

**Feasibility of getting $\mathcal{U}$.** Let $L = [y_1 x_1; y_2 x_2; \cdots; y_n x_n]$, then the column space of $L$ is close to $\mathcal{U}_m$ for $\gamma^\star = 0$, when $x_i$s have identity covariance. For $\gamma^\star = 0$, the standard top-$m$ SVD on $L$ in $\mathcal{O}(m^2 n)$ time with $\Omega\left(\frac{d}{\epsilon^2 \tau_{\min}^\star} \text{poly}\log(d)\right)$ samples is guaranteed to get a $\epsilon$-close estimate, following the well-known sin-theta theorem [7, 4]. For $\gamma^\star \neq 0$ under the same setting, we can use robust PCA methods to robustly find the subspace. For example, the state-of-the-art result in [8] provides a near optimal recovery guarantee, with slightly larger sample size (i.e., $\Omega(d^2/\epsilon^2)$). Closing this sample complexity gap is an interesting open problem for outlier robust PCA. Notice that instead of estimating the subspace, [14] uses the strong distributional assumption to calculate higher moments of Gaussian, and suffers from exponential dependency in $m$ in their sample complexity.

# 6 Discussion

Iterative least trimmed squares is the simplest instance of a much more general principle: that one can make learning robust to bad training data by iteratively updating a model using only the samples it best fits currently. In this paper we provide rigorous theoretical evidence that it obtains state-of-the-art results for a specific simple setting: mixed linear regression with corruptions. It is very interesting to see if this positive evidence can be established in other (and more general) settings.

While it seems similar at first glance, we note that our algorithm is not an instance of the Expectation-Maximization (EM) algorithm. In particular, it is tempting to associate a binary selection "hidden variable" $z_i$ for every sample $i$, and then use EM to minimize an overall loss that depends on $\theta$ and the $z$'s. However, this EM approach needs us to posit a model for the data under *both* the $z_i = 0$ (i.e. "discarded sample") and $z_i = 1$ (i.e. "chosen sample") choices. ILTS on the other hand only needs a model for the $z_i = 1$ case.

## Acknowledgement

We would like to acknowledge NSF grants 1302435 and 1564000 for supporting this research.

## Footnotes

[1]the component with fewest samples

[2]Notice that the larger $m$ becomes, the smaller the local region becomes, since $c_j$ depends on $m$. However, according to our bound for $\psi^+$ and $\psi^-$, the dependency of $c_j$ on $m$ is still polynomial.

[3]To satisfy this, one can always search through the set $\{1, c, c^2, c^3, \cdots\}$ (for some constant $c < 1$) and get an estimate in the interval $[c\tau_{(j)}^\star, \tau_{(j)}^\star)$.

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
