[Supplementary Material]

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

# A Supporting Lemmas

We give the key supporting lemmas in this section. Proof and discussions of these results are presented in Appendix E.

**Lemma 12.** *Let $\mathbf{A} \in \mathbb{R}^{n \times n}$ be a positive semi-definite matrix. $a, b \in \mathbb{R}^n$ are vectors such that $|a| < |b|$ element-wise. Then, there exists a diagonal matrix $\mathbf{N} \in \mathbb{R}^{n \times n}$, whose diaognal entries are either $1$ or $-1$, such that*

$$a^\top \mathbf{A} a < b^\top \mathbf{N} \mathbf{A} \mathbf{N} b.$$

**Lemma 13.** *For diagonal matrix $\mathbf{W}$, permutation matrix $\mathbf{P}$, diagonal matrix $\mathbf{N}$ with diagonal entries in $\{-1, 1\}$,*

$$\|\mathbf{X}^\top \mathbf{W} \mathbf{P} \mathbf{N} \mathbf{X}\|_2 \leq \max\{\|\mathbf{X}^\top \mathbf{W} \mathbf{X}\|_2, \|\mathbf{X}^\top \mathbf{N} \mathbf{P}^\top \mathbf{W} \mathbf{P} \mathbf{N} \mathbf{X}\|_2\}.$$

# B A Gradient Descent Variant of ILTS

In Algorithm 1, we find the least square solution for each round. Although this setting is more straightforward to analyze, exactly solving least square requires $d^3$ computation, while a gradient variant of finding an inexact solution may save computation in practice. This could be important since ILTS may be called for many times, as in Algorithm 2. In this part, we first analyze the gradient variant version of ILTS (GD-ILTS), and also give some guidance on achieving faster convergence speed using same number of gradient updates. Our gradient descent varaint of ILTS simply replaces **step 6** in Algorithm 1 by the sub-routine shown in Algorithm 3. We give the following result for the gradient variant of ILTS, in the exact MLR setting with $\gamma^\star = 0$ (for clearness). [4]

**Proposition 14** (ODE-based analysis for GD-ILTS). *Consider the gradient variant of Algorithm 1 using Algorithm 3 with infinitely small step size $\eta$ with $M_t = u$, under the same model setting as in Theorem 7 and same number of samples. Assume $Q, \tau$ being constant and $\gamma^\star = 0$, we have:*

$$\|\theta_{t+1} - \theta^\star\|_2 \leq \left\{ \frac{c_0 \lambda_t + 1/\nu(u)}{c_1 + 1/\nu(u)} + \omega(u) \right\} \|\theta_t - \theta^\star\|_2, \tag{5}$$

*where $\nu(u) = c_2 \left(e^{c_3 u} - 1\right)$, $\omega(u) \leq c_4 e^{-c_5 u}$, $\lambda_t = \left\{ \frac{\|\theta_t - \theta^\star_{(j)}\|_2}{\|\theta^\star_{(j)}\|_2} \vee \frac{\log n}{n} \right\}$.*

In (5), the smaller the $u$ is , the larger $\frac{c_0 \lambda_t + \frac{1}{\nu(u)}}{c_1 + \frac{1}{\nu(u)}}$ becomes, which will slow down the convergence.

Next, we analyze efficient number of gradient steps to take per round, based on Proposition 14. Let $w$ be the cost of one **step 5** in Algorithm 1. Define the approximate efficiency at round $t$ as follows, which measures the convergence rate with respect to the amount of computation:

$$\tilde{\mathcal{E}}(u; t, w) = \frac{\log \left( \frac{c_0 \lambda_t + 1/\nu(u)}{c_1 + 1/\nu(u)} \right)}{u + w}. \tag{6}$$

We ignore $\omega(u)$ in (5) since it is usually a small term, and makes the analysis difficult. We are interested in when $u$ achieves the maixmum for $\tilde{\mathcal{E}}(u; t, w)$. Notice that $\lambda_t$ changes with round number, i.e., when $\|\theta_t - \theta^\star\|_2$ gets to 0, $\lambda_t$ gets to zero. Our goal is to show given $\lambda_t$ (much smaller than $c_1$), how many gradient steps we need to take before moving to the next round.

**Proposition 15** (ideal stopping time for GD-ILTS). *Based on the approximate efficiency $\tilde{\mathcal{E}}(u; t, w)$ defined in (6), ILTS achieves its maximum guaranteed efficiency (approximately) by selecting $u \propto \log \frac{w}{\lambda_t \log \frac{1}{\lambda_t}}$, where $\lambda_t = \left\{ \frac{\|\theta_t - \theta^\star_{(j)}\|_2}{\|\theta^\star_{(j)}\|_2} \vee \frac{\log n}{n} \right\}$, and $w$ is the relative cost of **step 5** in ILTS.*

Proposition 15 implies that we should take number of gradient steps proportional to $\log \frac{1}{\|\theta_t - \theta^\star_{(j)}\|_2}$. Intuitively, as $\theta_t$ gets closer to $\theta^\star_{(j)}$, we should take more gradient steps, which is logarithmic in the inverse of current distance to the true component.

**Algorithm 3** Gradient descent variant for **step 6** in ILTS

1: $\theta_{t+1}^0 = \theta_t$
2: **for** $i = 0$ to $M_t - 1$ **do**
3: $\quad \theta_{t+1}^{i+1} \leftarrow \theta_{t+1}^i - \eta \frac{1}{2|S_t|} \sum_{j \in S_t} \nabla_\theta (y_i - \langle x_i, \theta \rangle)^2 |_{\theta = \theta_{t+1}^i}$
4: $\theta_{t+1} = \theta_{t+1}^T$

## C  Proofs in Section 4

### C.1  Proof of Theorem 4

*Proof.* We denote $\mathbf{W}_{(j)}^\star$, $\mathbf{W}_{(-j)}^\star$, $\mathbf{W}_R^\star$ to be the $n \times n$ diaognal $\{0,1\}$-matrix that represents $S_{(j)}^\star$, $S_{(-j)}^\star$, $R^\star$, respectively, for $j \in [m] \cup \{r\}$. Similarly, $\mathbf{W}_t$ is an $n \times n$ digonal $\{0,1\}$-matrix that represents the selected samples at round $t$ ($\mathbf{W}_{t,ii} = 1$ if sample $(x_i, y_i)$ is selected), which corresponds to $S_t$ in Algorithm 1.

**(I) Connect $\theta_{t+1}$ with $\theta_t$.** We consider the full update step: i.e., $\theta_{t+1} = (\mathbf{X}^\top \mathbf{W}_t \mathbf{X})^{-1} \mathbf{X}^\top \mathbf{W}_t y$. Following the notation in Section 3, we can rewrite $\theta_{t+1}$ as:

$$\theta_{t+1} = (\mathbf{X}^\top \mathbf{W}_t \mathbf{X})^{-1} \mathbf{X}^\top \mathbf{W}_t y$$

$$= (\mathbf{X}^\top \mathbf{W}_t \mathbf{X})^{-1} \mathbf{X}^\top \mathbf{W}_t \left( \mathbf{W}_{(j)}^\star \mathbf{X} \theta_{(j)}^\star + \sum_{l \in [m] \setminus \{j\}} \mathbf{W}_{(l)}^\star \mathbf{X} \theta_{(l)}^\star + \mathbf{W}_R^\star r \right)$$

$$= \theta_{(j)}^\star + (\mathbf{X}^\top \mathbf{W}_t \mathbf{X})^{-1} \left[ \mathbf{X}^\top \left( \mathbf{W}_t \mathbf{W}_{(j)}^\star - \mathbf{W}_t \right) \mathbf{X} \theta_{(j)}^\star + \sum_{l \in [m] \setminus \{j\}} \mathbf{X}^\top \mathbf{W}_t \mathbf{W}_{(l)}^\star \mathbf{X} \theta_{(l)}^\star + \mathbf{X}^\top \mathbf{W}_R^\star r \right]$$

$$= \theta_{(j)}^\star + (\mathbf{X}^\top \mathbf{W}_t \mathbf{X})^{-1} \left[ \sum_{l \in [m] \setminus \{j\}} \mathbf{X}^\top \mathbf{W}_t \mathbf{W}_{(l)}^\star \mathbf{X} \left( \theta_{(l)}^\star - \theta_{(j)}^\star \right) - \mathbf{X}^\top \mathbf{W}_t \mathbf{W}_R^\star \left( \mathbf{X} \theta_{(j)}^\star - r \right) \right],$$

where we used the fact that $\mathbf{W}_t - \mathbf{W}_t \mathbf{W}_{(j)}^\star = \sum_{l \in [m] \setminus \{j\}} \mathbf{W}_t \mathbf{W}_{(l)}^\star + \mathbf{W}_t \mathbf{W}_{-1}^\star$. As a result, the $\ell_2$ distance can be bounded:

$$\|\theta_{t+1} - \theta_{(j)}^\star\|_2 \leq \frac{1}{\sigma_{\min}(\mathbf{X}^\top \mathbf{W}_t \mathbf{X})} \underbrace{\left\| \sum_{l \in [m] \setminus \{j\}} \mathbf{X}^\top \mathbf{W}_t \mathbf{W}_{(l)}^\star \mathbf{X}(\theta_{(l)}^\star - \theta_{(j)}^\star) - \mathbf{X}^\top \mathbf{W}_t \mathbf{W}_R^\star (\mathbf{X}\theta_{(j)}^\star - r) \right\|_2}_{\mathcal{T}_1}.$$

By triangle inequality,

$$\mathcal{T}_1 \leq \underbrace{\left\| \mathbf{X}^\top \mathbf{W}_t \left[ \sum_{l \in [m] \setminus \{j\}} \left( \mathbf{W}_{(l)}^\star \mathbf{X}(\theta_{(l)}^\star - \theta_t) \right) + \mathbf{W}_R^\star (r - \mathbf{X}\theta_t) \right] \right\|_2}_{\mathcal{T}_2} + \underbrace{\left\| \mathbf{X}^\top \mathbf{W}_t \left( \mathbf{W}_{(-j)}^\star + \mathbf{W}_R^\star \right) \mathbf{X}(\theta_t - \theta_{(j)}^\star) \right\|_2}_{\mathcal{T}_3}.$$

**(II) Map between errors.** We next focus on $\mathcal{T}_2$. Notice that $S_t \cap S_{(l)}^\star$ is the set of samples selected by the algorithm, which means that they have smaller values in $|x_i^\top (\theta_{(l)}^\star - \theta_t)|$, $\forall l \in [m] \setminus \{j\}$. Similarly, samples in set $S_t \cap R^\star$ also have small values in $|r_i - x_i^\top \theta_t|$. Also, since $|S_t| < |S_{(j)}^\star|$, we always have more samples from $S_{(j)}^\star$ that are not selected (due to larger losses) than samples in $S_t \cap \left( \cup_{l \in [m] \setminus \{j\}} S_{(l)}^\star \cup R^\star \right)$. Therefore, there exists a permutation matrix $\mathbf{P} \in \mathbb{R}^{n \times n}$, such that (inequality holds element-wise):

$$\left| \mathbf{W}_t \left[ \sum_{l \in [m] \setminus \{j\}} \left( \mathbf{W}_{(l)}^\star \mathbf{X}(\theta_{(l)}^\star - \theta_t) \right) + \mathbf{W}_R^\star (r - \mathbf{X}\theta_t) \right] \right| \leq \left| \mathbf{W}_t \left( \mathbf{W}_{(-j)}^\star + \mathbf{W}_R^\star \right) \mathbf{P} \mathbf{X}(\theta_{(j)}^\star - \theta_t) \right|.$$

Let us denote

$$a = \mathbf{W}_t \left[ \sum_{l \in [m] \setminus \{j\}} \left( \mathbf{W}_{(l)}^\star \mathbf{X}(\theta_{(l)}^\star - \theta_t) \right) + \mathbf{W}_R^\star \left( r - \mathbf{X}\theta_t \right) \right], b(\mathbf{N}) = \mathbf{W}_t \left( \mathbf{W}_{(-j)}^\star + \mathbf{W}_R^\star \right) \mathbf{P} \mathbf{N} \mathbf{X}(\theta_{(j)}^\star - \theta_t).$$

According to Lemma 12, there exists a diagonal matrix $\mathbf{N}$, such that $\|\mathbf{X}^\top a\|_2 \leq \|\mathbf{X}^\top b(\mathbf{N})\|_2$.

**(III) Plug-in feature regularity property and affine error property.** According to Lemma 13, we know

$$\mathcal{T}_2 \leq \left\| \mathbf{X}^\top \mathbf{W}_t \left( \mathbf{W}_{(-j)}^\star + \mathbf{W}_R^\star \right) \mathbf{P} \mathbf{N} \mathbf{X}(\theta_{(j)}^\star - \theta_t) \right\|_2 \leq \left\| \mathbf{X}^\top \mathbf{W}_t \left( \mathbf{W}_{(-j)}^\star + \mathbf{W}_R^\star \right) \mathbf{P} \mathbf{N} \mathbf{X} \right\|_2 \left\| \theta_{(j)}^\star - \theta_t \right\|_2$$

$$\leq \max \left\{ \left\| \mathbf{X}^\top \mathbf{W}_t \left( \mathbf{W}_{(-j)}^\star + \mathbf{W}_R^\star \right) \mathbf{X} \right\|_2, \left\| \mathbf{X}^\top \mathbf{N} \mathbf{P}^\top \mathbf{W}_t \left( \mathbf{W}_{(-j)}^\star + \mathbf{W}_R^\star \right) \mathbf{P} \mathbf{N} \mathbf{X} \right\|_2 \right\} \left\| \theta_{(j)}^\star - \theta_t \right\|_2.$$

By feature regularity property in Definition 2, both $\mathcal{T}_2$ and $\mathcal{T}_3$ are less than $\psi^+(|S_t \cap (S_{(-j)}^\star \cup R^\star)|)\|\theta_t - \theta_{(j)}^\star\|_2$. As a result,

$$\left\| \theta_{t+1} - \theta_{(j)}^\star \right\|_2 \leq \frac{2\psi^+ \left( \left| S_t \cap \left( S_{(-j)}^\star \cup R^\star \right) \right| \right)}{\psi^-(\alpha n)} \left\| \theta_t - \theta_{(j)}^\star \right\|_2.$$

Finally, notice that

$$S_t \cap \left( S_{(-j)}^\star \cup R^\star \right) = S_t \cap S_{(-j)}^\star + S_t \cap R^\star.$$

Here,

$$|S_t \cap R^\star| \leq |R^\star| = \gamma^\star \tau_{\min}^\star n.$$

On the other hand, by Definition 3, $|S_t \cap S_{(-j)}^\star| \leq \mathcal{V}(\Delta)$ for

$$\Delta = \frac{\left\| \theta_t - \theta_{(j)}^\star \right\|_2}{\min_{l \in [m] \setminus \{j\}} \left\| \theta_t - \theta_{(l)}^\star \right\|_2}$$

$$\leq \frac{\left\| \theta_t - \theta_{(j)}^\star \right\|_2}{\min_{l \in [m] \setminus \{j\}} \left\| \theta_{(j)}^\star - \theta_{(l)}^\star \right\|_2 - \left\| \theta_{(j)}^\star - \theta_t \right\|_2}$$

$$\leq \frac{2 \left\| \theta_t - \theta_{(j)}^\star \right\|_2}{\min_{l \in [m] \setminus \{j\}} \left\| \theta_{(j)}^\star - \theta_{(l)}^\star \right\|_2}$$

$$= \frac{1}{Q_j} \cdot \frac{2 \left\| \theta_t - \theta_{(j)}^\star \right\|_2}{\left\| \theta_{(j)}^\star \right\|_2},$$

where the last inequality uses the property that $\|\theta_t - \theta_{(j)}^\star\|_2 \leq \frac{1}{2} \min_{l \in [m] \setminus \{j\}} \|\theta_{(j)}^\star - \theta_{(l)}^\star\|_2$, and the last equality uses Definition 1.

Combining all the results, we have:

$$\left\| \theta_{t+1} - \theta_{(j)}^\star \right\|_2 \leq \frac{2\psi^+ \left( \mathcal{V} \left( \frac{1}{Q_j} \cdot \frac{2\|\theta_t - \theta_{(j)}^\star\|_2}{\|\theta_{(j)}^\star\|_2} \right) + \gamma^\star \tau_{\min}^\star n \right)}{\psi^-(\tau n)} \left\| \theta_t - \theta_{(j)}^\star \right\|_2. \tag{7}$$

∎

## C.2  Proof of Lemma 5

*Proof.* We notice that [2] provides a bound for the same setting. In terms of our notation, their results show that with probability $1 - \delta$,

$$\psi^+(k) \leq k \left( 1 + 3e\sqrt{6 \log \frac{en}{k}} \right) + \mathcal{O} \left( \sqrt{np + n \log \frac{1}{\delta}} \right), \tag{8}$$

$$\psi^-(k) \geq n - (n - k) \left( 1 + 3e\sqrt{6 \log \frac{en}{n-k}} \right) - \Omega \left( \sqrt{np + n \log \frac{1}{\delta}} \right). \tag{9}$$

Their result (8) directly gives us the desired bound for $\psi^+(k)$, i.e., $c_{c_k,1} = 1 + 3e\sqrt{6\log\frac{2}{c_k}} + \frac{c_1}{c_k}$, where $c_1$ comes from $\mathcal{O}(\sqrt{nd + n\log\frac{1}{\delta}})$, and $\delta = e^{-c_2 n}$.

On the other hand, the bound on $\psi^-(k)$ in (9) is only meaningful for a large $k$. For example, for $k = 0.1n$, it is easy to check that the RHS of (9) is negative, no matter how large $n$ is. The reason is due to their proof technique. More specifically, they take uniform bound over all possible $\mathbf{W}$s, the size of which is exponential in $n$ (for $k = c_k n$). This makes their uniform bound weak, and hence the lower tail bound would not hold for $k$s with small constant $c_k$.

Here, we take another approach: we bound the quantity $\psi^-(k)$ by taking an $\epsilon$-net over the parameter space (with ambient dimension $d$). Notice that although the size of this net is large, it is exponential in the dimension $d$ (not in $n$), and by using uniform bound, we can take a large enough $n$ to absorb all tails into a small tail. More specifically, let $\Theta_\epsilon$ be an $\epsilon$-net cover the unit $d$-dimensional sphere. Then, for any unit norm vector $\tilde{\theta}$, their exists some $\theta \in \Theta_\epsilon$ close to $\tilde{\theta}$,

$$\sqrt{\tilde{\theta}^\top \mathbf{X}^\top \mathbf{W} \mathbf{X} \tilde{\theta}} = \sqrt{(\tilde{\theta} - \theta + \theta)^\top \mathbf{X}^\top \mathbf{W} \mathbf{X} (\tilde{\theta} - \theta + \theta)} \geq \sqrt{\theta^\top \mathbf{X}^\top \mathbf{W} \mathbf{X} \theta} - \epsilon\sqrt{\psi^+(k)}. \quad (10)$$

Notice that for any fixed $\theta$, $\theta^\top \mathbf{X}^\top \mathbf{W} \mathbf{X} \theta$ is a subset of sum of squares of Gaussian random variables. In another word,

$$\min_{\mathbf{W} \in \mathcal{W}_k} \theta^\top \mathbf{X}^\top \mathbf{W} \mathbf{X} \theta \geq \sum_{i=1}^{k} r_{(i)}, r_i = (\mathbf{x}_i^\top \theta)^2,$$

where $r_{(i)}$ is the $i$-th smallest value in vector $r$. Notice that [3] showed that the quantile has sub-gamma property. Following their results in Section 4, we have

$$\Pr\left[r_{(\frac{k}{2})} \leq F_r^{-1}\left(\frac{k}{2n}\right) - c_3\right] \leq e^{-c_4 c_k n},$$

where $F_r$ is the cumulative distribution function of $r$, with randomness coming from $x_i$s. Notice that the size of the set $\Theta_\epsilon$ is upper bounded by $\left(\frac{3}{\epsilon}\right)^d$. Then, with probability $1 - e^{-c_4 c_k n + d\log\frac{3}{\epsilon}}$, $\theta^\top \mathbf{X}^\top \mathbf{W} \mathbf{X} \theta \geq c_5 F_r^{-1}(\frac{c_k}{2}) c_k n \geq c_6 c_k^2 n$. By selecting $\epsilon$ as a small constant ($\frac{c_7 c_k^2}{1 + c_8\sqrt{\log\frac{1}{c_k}} + \frac{c_9}{c_k}}$, based on the bounds for both $\psi^+$ with arbitrary direction and $\psi^-$ for fixed direction), and $n > \frac{1}{2c_4 c_k} d\log\frac{3}{\epsilon}$ (in order to make uniform bound of the tails over $\epsilon$-net small), according to (10), with high probability $1 - e^{-\frac{c_4 c_k}{2} n}$, $\psi^-(k) \geq c_{c_k,2} c_k n$, for $c_{c_k,2} \approx c_{10} c_k$. ∎

Notice that similar results for $\psi^+$ and $\psi^-$ in (1) hold for sub-Gaussian random variables with bounded condition number. Lemma 16 and Theorem 17 in [2] gives a guarantee when $k$ is comparatively large. Following our proof technique, in order to show similar properties for smaller $k$, we require concentration of order statistics, which also holds true for a wide class of sub-Gaussian distributions, which is dicussed in [3]. As a special case, Lemma 5 can be easily generalized to the setting where $x_i \sim \mathcal{N}(0, \Sigma_{(j)}), \forall i \in S_{(j)}^\star$, with $\mathbf{I} \preceq \Sigma_{(j)} \preceq \sigma\mathbf{I}$. This is given in Lemma 16.

### C.3 Proof of Lemma 6

*Proof.* Recall the definition in Definition 3, $V(\Delta)$ is the maximum number of affine error one can make on any affine directions $v_1, v_2$. For simplicity, we assume $\|v_2\|_2 = 1$, hence $\|v_1\|_2 = \Delta$. We fist study the result for all fixed $v_1, v_2$ in $d$-dimension such that $\|v_1\|_2 = \Delta, \|v_2\|_2 = 1$. We show a high probability upper bound on $\mathcal{V}(\Delta)$ for any fixed $v_1, v_2$. Then, by using $\epsilon$-net argument, we provide the upper bound for arbitrary $v_1, v_2$s.

**(I)** Given fixed $v_1, v_2$, studying $\mathcal{V}(\Delta)$ can be reduced to the following problem: Suppose we have two Gaussian distributions $\mathcal{D}_1 = \mathcal{N}(0, \Delta^2), \mathcal{D}_2 = \mathcal{N}(0, 1)$. We have $\tau_1^\star n$ i.i.d. samples from $\mathcal{D}_1$ and $\tau_2^\star n$ i.i.d. samples from $\mathcal{D}_2$. Denote the set of the top $\tau_1^\star n$ samples with smallest abstract values as $S_{\tau_1^\star n}$. Then, for $\Delta \leq 1$, what is the upper bound on the number of samples in $S_{\tau_1^\star n}$ that are from $\mathcal{D}_2$?

Let $S_1^\star, S_2^\star$ be the set of samples from $\mathcal{D}_1, \mathcal{D}_2$, respectively, and let $S_1 := S_{\tau_1^\star n}$. Consider $|S_1 \cap S_2^\star|$, by definition, let $\delta$ be the threshold between samples in $S_1 \cap S_1^\star$ and samples in $S_1^\star \backslash S_1$ . Since

there are at least $(1 - c_\tau)\tau_1^\star n$ samples in $S_1^\star$ that are not in $S_1$, by the sub-Gamma property of order statistics of Gaussian random variables [3], we know

$$\Pr\left[\delta > F_\Delta^{-1}(c_\tau) + c_0\Delta\right] \leq e^{-c_1\tau_1^\star n}. \tag{11}$$

As a result, $\delta \leq c_2\Delta$ with high probability.

**(II)** On the other hand, for a random variable $u_2 \sim \mathcal{D}_2$, we know that $\Pr[|u_2| \leq \delta] \leq \sqrt{\frac{2}{\pi}}\delta$, which is tight for small $\delta$. Let $\mathcal{M}_{\delta,i}$ be the event *sample $u_i$ from $\mathcal{D}_2$ has abstract value less than $\delta$*, and a Bernoulli random variable $m_{i,\delta}$ that is the indicator of event $\mathcal{M}_{\delta,i}$ holds or not. Then,

$$\mathbb{E}\left[\sum_{i=1}^{\tau_2^\star n} m_{i,\delta}\right] \leq \sqrt{\frac{2}{\pi}}\delta\tau_2^\star n.$$

For independent Bernoulli random variable $x_i$s, $i \in [\tilde{n}]$ with $X = \sum_i x_i$ and $\mu = \mathbb{E}[X]$, Chernoff's inequality gives [24]

$$\Pr[X \geq t] \leq e^{-t}$$

for any $t \geq e^2\tilde{n}\mu$. In the above setting we consider, we have with high probability $1 - n^{-c}$, $\sum_{i=1}^{\tau_2^\star n} m_{i,\delta} \leq c\max\{\tau_2^\star n\Delta, \log n\}$.

Next, we use an $\epsilon$-net argument to prove for arbitrary $v_1, v_2$ in **Part I**. Notice that we select $\epsilon = \frac{\Delta}{\sqrt{\log n}}$, take uniform bound over all fixed vectors, and require $n \geq \frac{C}{\tau_1^\star}d\log\log d$. Then, with probability $1 - n^{-c}$, the threshold on any direction $v_1, v_2$ satisfies $\tilde{\delta} < \delta + c\Delta$.

In summary, we have $\mathcal{V}(\Delta) \leq c\{\Delta n \vee \log n\}$ as long as $n \geq c\frac{d\log\log d}{\min_{j \in [m]}\tau_{(j)}^\star}$.

∎

### C.4 Proof of Theorem 7

*Proof.* According to Theorem 4,

$$\left\|\theta_{t+1} - \theta_{(j)}^\star\right\|_2 \leq \frac{2\psi^+\left(\mathcal{V}\left(\frac{1}{Q_j} \cdot \frac{2\|\theta_t - \theta_{(j)}^\star\|_2}{\|\theta_{(j)}^\star\|_2}\right) + \gamma^\star\tau_{\min}^\star n\right)}{\psi^-(\tau n)}\left\|\theta_t - \theta_{(j)}^\star\right\|_2. \tag{12}$$

Then, according to Lemma 6,

$$\mathcal{V}\left(\frac{1}{Q_j} \cdot \frac{2\left\|\theta_t - \theta_{(j)}^\star\right\|_2}{\left\|\theta_{(j)}^\star\right\|_2}\right) \leq c\left\{\frac{1}{Q_j} \cdot \frac{2\left\|\theta_t - \theta_{(j)}^\star\right\|_2}{\left\|\theta_{(j)}^\star\right\|_2}n \vee \log n\right\},$$

and based on the results from Lemma 5, we have:

$$\frac{2\psi^+\left(\mathcal{V}\left(\frac{1}{Q_j} \cdot \frac{2\|\theta_t - \theta_{(j)}^\star\|_2}{\|\theta_{(j)}^\star\|_2}\right) + \gamma^\star\tau_{\min}^\star n\right)}{\psi^-(\tau n)}$$

$$\leq \frac{2c_{c_\tau,1}\left\{c\left\{\frac{1}{Q_j} \cdot \frac{2\|\theta_t - \theta_{(j)}^\star\|_2}{\|\theta_{(j)}^\star\|_2}n \vee \log n\right\} + \gamma^\star\tau_{\min}^\star n\right\}}{c_{c_\tau,2}\tau n}$$

$$\leq c_0\frac{c_{c_\tau,1}\left(\frac{1}{Q_j} \cdot \frac{2\|\theta_t - \theta_{(j)}^\star\|_2}{\|\theta_{(j)}^\star\|_2} + \gamma^\star\tau_{\min}^\star\right)}{c_{c_\tau,2}\tau}.$$

In order to guarantee that $\theta_{t+1}$ is getting closer, we require:

$$c_0 \frac{c_{c_\tau,1}\left(\frac{1}{Q_j}\cdot\frac{2\|\theta_t-\theta^\star_{(j)}\|_2}{\|\theta^\star_{(j)}\|_2}+\gamma^\star\tau^\star_{\min}\right)}{c_{c_\tau,2}\tau}\leq\frac{1}{2}$$

$$\Rightarrow\left\|\theta_t-\theta^\star_{(j)}\right\|_2\leq c_1\left(\frac{c_{c_\tau,2}\tau}{c_{c_\tau,1}}-\gamma^\star-\tau^\star_{\min}\right)Q_j\left\|\theta^\star_{(j)}\right\|_2=c_1\left(\frac{c_{c_\tau,2}\tau}{c_{c_\tau,1}}-\gamma^\star-\tau^\star_{\min}\right)\min_{l\in[m]\setminus\{j\}}\left\|\theta^\star_{(j)}-\theta^\star_{(l)}\right\|_2.$$

As long as $\gamma^\star\leq\frac{c_{c_\tau,2}\tau}{2c_{c_\tau,1}\tau^\star_{\min}}$, we only require the following sufficient condition for $\theta_t$:

$$\left\|\theta_t-\theta^\star_{(j)}\right\|_2\leq c_2\frac{c_{c_\tau,2}\tau}{c_{c_\tau,1}}\min_{l\in[m]\setminus\{j\}}\left\|\theta^\star_{(j)}-\theta^\star_{(l)}\right\|_2,$$

and the required sample complexity is based on Lemma 5 and Lemma 6. ∎

## C.5 Proof of Corollary 8

The result of Corollary 8 is mostly built upon the result in Theorem 7. Instead, we use similar results for the feature regularity property and affine error property, as given in Lemma 5 and Lemma 6. For the affine error property, $\mathcal{V}(\Delta)$ only changes by a multiplicative factor of $\sigma$, i.e., $\mathcal{V}^{\texttt{new}}(\Delta)\leq\mathcal{V}(\sigma\Delta)$ which is straightforward to see. For the feature regulairty property, we use Lemma 16.

# D  Proof in Section 5

## D.1  Proof of Theorem 11

*Proof.* In Theorem 7, we show that ILTS locally converges as long as the intialization satisfies:

$$\|\theta_t-\theta^\star_{(j)}\|_2\leq c_2\frac{c_{c_\tau,2}\tau}{c_{c_\tau,1}}\min_{l\in[m]\setminus\{j\}}\|\theta^\star_{(j)}-\theta^\star_{(l)}\|_2.$$

Given an $\epsilon$-close subspace $\mathcal{U}$, consider a $\tilde{m}$-dimenional sphere with radius $\max_{j\in[m]}\|\theta^\star_{(j)}\|_2$, and an $\epsilon$-net over this sphere with $\epsilon=\frac{c_2}{2}\frac{c_{c_\tau,2}\tau}{c_{c_\tau,1}}\min_{l\in[m]\setminus\{j\}}\|\theta^\star_{(j)}-\theta^\star_{(l)}\|_2$. Then, we know the size of this $\epsilon$-net is upper bounded by [24]

$$\left(\frac{3\max_{j\in[m]}\|\theta^\star_{(j)}\|_2}{\frac{c_2}{2}\frac{c_{c_\tau,2}\tau}{c_{c_\tau,1}}\min_{l\in[m]\setminus\{j\}}\|\theta^\star_{(j)}-\theta^\star_{(l)}\|_2}\right)^{\tilde{m}}\leq\left(\frac{c_3c_{c_\tau,1}}{c_{c_\tau,2}\tau Q}\right)^{\tilde{m}}=\left(\frac{1}{\text{poly}(\tau)Q}\right)^{\mathcal{O}(m)}=\left(\frac{1}{\tau Q}\right)^{\mathcal{O}(m)}.$$

Also, there always exists a vector $\theta_\epsilon$ in this $\epsilon$-net, which is $\epsilon$-close to the projection of $\theta^\star_{(j)}$ to $\mathcal{U}$ (denoted as $\mathcal{U}(\theta_{(j)})$). Therefore,

$$\|\theta_\epsilon-\theta^\star_{(j)}\|_2\leq\|\theta_\epsilon-\mathcal{U}(\theta_{(j)})\|_2+\|\mathcal{U}(\theta_{(j)})-\theta^\star_{(j)}\|_2\leq\epsilon+\epsilon=c_2\frac{c_{c_\tau,2}\tau}{c_{c_\tau,1}}\min_{l\in[m]\setminus\{j\}}\|\theta^\star_{(j)}-\theta^\star_{(l)}\|_2.$$

With $\left(\frac{1}{\tau Q}\right)^{\mathcal{O}(m)}$ of initializations, there always exist an initialization such that ILTS will succefully recover a single component. Therefore, given this $\epsilon$-close subspace, with $n=\Omega\left(\frac{d\log\log d}{\tau^\star_{\min}}\right)$ samples and in time $\left(\frac{1}{\tau Q}\right)^{\mathcal{O}(m)}nd^2\log\frac{1}{\varepsilon}$.

∎

# E  Proofs in Section A

## E.1  Proof of Lemma 12

*Proof.* The proof is straightforward. We know that

$$\tilde{b}^\top\mathbf{A}\tilde{b}=\left(a+\tilde{b}-a\right)^\top\mathbf{A}\left(a+\tilde{b}-a\right)=a^\top\mathbf{A}a+(\tilde{b}-a)^\top\mathbf{A}(\tilde{b}-a)+2(\tilde{b}-a)^\top\mathbf{A}a.$$

Therefore, as long as $(\tilde{b} - a)^\top \mathbf{A} a > 0$, we have $a^\top \mathbf{A} a \leq \tilde{b}^\top \mathbf{N} \mathbf{A} \mathbf{N} \tilde{b}$. Now, denote $\tilde{a} := \mathbf{A} a$. We choose $\mathbf{N}$ such that $\texttt{sgn}(\mathbf{N}_{ii} b_i) = \texttt{sgn}(\tilde{a}), \forall i \, in \, [n]$, and let $\tilde{b} = \mathbf{N} b$. Notice that since $|b_i| > |a_i|$ element-wise, $\texttt{sgn}(\tilde{b}) = \texttt{sgn}(\tilde{b} - a)$. Therefore, the inner product between $\tilde{b} - a$ and $\mathbf{A} a$ is always positive since each entry in both vectors is either both positive or both negative. ∎

## E.2 Proof of Lemma 13

*Proof.* This result is based on the spectral norm inequality $\|\mathbf{AB}\|_2 \leq \|\mathbf{A}\|_2 \|\mathbf{B}\|_2$, and as a result, $\|\mathbf{AB}\|_2 \leq \max\{\|\mathbf{A}\|_2^2, \|\mathbf{B}\|_2^2\}$. On the other hand, it is easy to check by definition:

$$\|\mathbf{X}^\top \mathbf{W} \mathbf{P} \mathbf{N} \mathbf{X}\|_2 = \max_{u,v: \|u\|_2 = \|v\|_2 = 1} u^\top \mathbf{X}^\top \mathbf{W} \mathbf{P} \mathbf{N} \mathbf{X} v.$$

For convenience, let $u, v$ be the unit $d$ dimensional vectors that achieve the maximum. Let $\tilde{u} = \mathbf{X} u, \tilde{v} = \mathbf{X} v$. Then, the RHS of the above equation is upper bounded by

$$\sum_{i \in \text{Tr}(\mathbf{W})} |\tilde{u}_{s_{i,1}} \tilde{v}_{s_{i,2}}| \leq \sqrt{\left( \sum_{i \in \text{Tr}(\mathbf{W})} \tilde{u}_{s_{i,1}}^2 \right) \left( \sum_{i \in \text{Tr}(\mathbf{W})} \tilde{v}_{s_{i,2}}^2 \right)}$$

$$\leq \max \left\{ \sum_{i \in \text{Tr}(\mathbf{W})} \tilde{u}_{s_{i,1}}^2, \sum_{i \in \text{Tr}(\mathbf{W})} \tilde{v}_{s_{i,2}}^2 \right\}$$

$$= \max \left\{ u^\top \mathbf{X}^\top \mathbf{W} \mathbf{X} u, v^\top \mathbf{X}^\top \mathbf{N} \mathbf{P}^\top \mathbf{W} \mathbf{P} \mathbf{N} \mathbf{X} v \right\}$$

$$\leq \max \left\{ \left\| \mathbf{X}^\top \mathbf{W} \mathbf{X} \right\|_2, \left\| \mathbf{X}^\top \mathbf{N} \mathbf{P}^\top \mathbf{W} \mathbf{P} \mathbf{N} \mathbf{X} \right\|_2 \right\},$$

where $s_{i,1}$s and $s_{i,2}$s are two index sequences. ∎

## E.3 Restricted Subset Property for More General Distributions

**Lemma 16** (non-isotropic Gaussian distributions). *Let $\psi^+(k), \psi^-(k)$ be defined as in (1), assume each $x_i \sim \mathcal{N}(0, \Sigma_{(j)})$ for $i \in S_{(j)}^\star$, $\mathbf{I} \preceq \Sigma_{(j)} \preceq \sigma \mathbf{I}$. Then, for $k = c_k n$ with constant $c_k$, for $n = \Omega \left( \frac{d \log \frac{\sigma}{c_k}}{c_k} \right)$, with high probability,*

$$\psi^+(k) \leq c_{c_k,1} \cdot \sigma k, \psi^-(k) \geq c_{c_k,2} \cdot k.$$

*Proof.* The proof is similar to the proof of Lemma 5. For $\psi^+(k)$, we can simply bound it by a mulitplicative factor of $\sigma$. For $\psi^-(k)$, according to (10), we require an additional $\log \sigma$ factor for $n$, since a finer net with $\tilde{\epsilon} = \frac{\epsilon}{\sqrt{\sigma}}$ is needed. ∎

# F Proofs in Section B

## F.1 Proof of Proposition 14

*Proof.* We connect the updated parameter at each epoch with a closed form solution to a penalized minimization problem. More specifically, accordng to [22], define

$$\dot{\theta}(t) := \frac{d}{dt} \theta(t) = -\nabla f(\theta(t)), \theta(0) = \theta_0,$$

and

$$\underline{\theta}(\nu) = \arg \min_\theta f(\theta) + \frac{1}{2\nu} \|\theta - \theta_0\|_2^2,$$

where $f(\theta) = \frac{1}{2|S|} \sum_{i \in S} (y_i - x_i^\top \theta)^2$. Then, $\theta(t)$ and $\underline{\theta}(\nu)$ have the following relationship:

$$\|\theta(t) - \underline{\theta}(\nu(t))\|_2 \leq \frac{\|\nabla f(\theta_0)\|_2}{m} \left( e^{-mt} + \frac{c_{\text{ode}}}{1 - c_{\text{ode}} - e^{c_{\text{ode}} M t}} \right),$$

where $\nu(t) = \frac{1}{c_{\text{ode}}m}\left(e^{c_{\text{ode}}Mt} - 1\right)$, for $m = \frac{1}{|S|}\sigma_{\min}(\mathbf{X}^\top\mathbf{W}\mathbf{X})$, $M = \frac{1}{|S|}\sigma_{\max}(\mathbf{X}^\top\mathbf{W}\mathbf{X})$, $c_{\text{ode}} = \frac{2m}{M+m}$. Since $\underline{\theta}(\nu)$ has a closed form solution in this linear setting, by connecting $\theta^{t+1}$ with $\underline{\theta}$, we are able to bound $\theta^{t+1}$ using similar proof technique as above.

$$\underline{\theta}(\nu) = \arg\min_\theta \underbrace{\frac{1}{2}\frac{1}{|S|}(y_S - \mathbf{X}_S\theta)^\top (y_S - \mathbf{X}_S\theta) + \frac{1}{2\nu}\|\theta - \theta_0\|_2^2}_{L(\boldsymbol{\theta})}.$$

Observe that for $\underline{\theta}(\nu)$ satisfies first order condition:

$$\nabla L(\theta) = \frac{1}{|S|}\mathbf{X}_S^\top (\mathbf{X}_S\theta - y_S) + \frac{1}{\nu}(\theta - \theta_0), \nabla L(\underline{\theta}(\nu)) = 0,$$

which gives the following closed form solution:

$$\underline{\theta}(\nu) = \left(\frac{1}{|S|}\mathbf{X}_S^\top\mathbf{X}_S + \frac{1}{\nu}\mathbf{I}\right)^{-1}\left(\frac{1}{|S|}\mathbf{X}_S^\top y_S + \frac{1}{\nu}\theta_0\right)$$

$$= \left(\frac{1}{\text{Tr}(\mathbf{W})}\mathbf{X}^\top\mathbf{W}\mathbf{X} + \frac{1}{\nu}\mathbf{I}\right)^{-1}\left(\frac{1}{\text{Tr}(\mathbf{W})}\mathbf{X}^\top\mathbf{W}\left(\mathbf{W}_{(j)}^\star\mathbf{X}\theta^\star + \sum_{l\in[m]\setminus\{j\}}\mathbf{W}_{(l)}^\star\mathbf{X}\theta_{(l)}^\star + \mathbf{W}_R^\star r\right) + \frac{1}{\nu}\theta_0\right)$$

$$= \theta^\star + \left(\frac{1}{\text{Tr}(\mathbf{W})}\mathbf{X}^\top\mathbf{W}\mathbf{X} + \frac{1}{\nu}\mathbf{I}\right)^{-1}\left(-\frac{1}{\text{Tr}(\mathbf{W})}\mathbf{X}^\top\mathbf{W}\mathbf{W}_{(-1)}^\star(\mathbf{X}\theta^\star - r) + \frac{1}{\nu}(\theta_0 - \theta^\star)\right)$$

$$+ \sum_{l\in[m]\setminus\{j\}}\left(\frac{1}{\text{Tr}(\mathbf{W})}\mathbf{X}^\top\mathbf{W}\mathbf{X} + \frac{1}{\nu}\mathbf{I}\right)^{-1}\frac{1}{\text{Tr}(\mathbf{W})}\mathbf{X}^\top\mathbf{W}\mathbf{W}_{(l)}^\star\mathbf{X}(\theta_{(l)}^\star - \theta_{(j)}^\star).$$

Based on the same proof technique as in Theorem 7, we have

$$\|\underline{\theta}(\nu) - \theta^\star\|_2 \le \frac{\frac{c_{c_\tau,1}}{\tau}\left\{c\left\{\frac{1}{Q_j}\cdot\frac{2\|\theta_t - \theta_{(j)}^\star\|_2}{\|\theta_{(j)}^\star\|_2} \vee \frac{\log n}{n}\right\} + \gamma^\star\tau_{\min}^\star\right\} + \frac{1}{\nu}}{c_{c_\tau,2} + \frac{1}{\nu}}\|\theta_0 - \theta^\star\|_2. \qquad (13)$$

On the other hand,

$$\|\underline{\theta}(\nu) - \theta(t)\|_2 \le \frac{\|\nabla f(\theta_0)\|_2}{m}\left(e^{-mt} + \frac{c_{\text{ode}}}{1 - c_{\text{ode}} - e^{c_{\text{ode}}Mt}}\right) \qquad (14)$$

$$\le \frac{\psi^+(\tau n)}{\psi^-(\tau n)}\left(e^{-mt} + \frac{c_{\text{ode}}}{1 - c_{\text{ode}} - e^{c_{\text{ode}}Mt}}\right)\|\theta_0 - \theta^\star\|_2. \qquad (15)$$

Combining (13) and (15), setting $\theta_{t+1} = \theta(t)$, $\theta_t = \theta_0$, we have:

$$\|\theta(t) - \theta^\star\|_2 \le \left\{\frac{\frac{c_{c_\tau,1}}{\tau}\left\{c\left\{\frac{1}{Q_j}\cdot\frac{2\|\theta_t - \theta_{(j)}^\star\|_2}{\|\theta_{(j)}^\star\|_2} \vee \frac{\log n}{n}\right\} + \gamma^\star\tau_{\min}^\star\right\} + \frac{1}{\nu}}{c_{c_\tau,2} + \frac{1}{\nu}} + \omega(c_\tau, m, M, c_{\text{ode}})\right\}\|\theta_0 - \theta^\star\|_2,$$

$$(16)$$

where $\omega(c_\tau, m, M, c_{\text{ode}}) = \frac{c_{c_\tau,1}}{c_{c_\tau,2}}\left(e^{-mt} + \frac{c_{\text{ode}}}{1 - c_{\text{ode}} - e^{c_{\text{ode}}Mt}}\right)$.

∎

## F.2 Proof of Proposition 15

*Proof.* Consider the result in Proposition 5, let $\mathcal{C}_1 = c_1$, $\mathcal{C}_2(t) = c_0\lambda_t$. Then, $\|\theta_{t+1} - \theta^\star\|_2 \le \left(\frac{\mathcal{C}_2(t) + \frac{1}{\nu(u)}}{\mathcal{C}_1 + \frac{1}{\nu(u)}} + \omega(u)\right)\|\theta_t - \theta^\star\|_2$. The following result writes for $m = M = 1$, for simplicty, since $m, M$ are both constants. Our goal is to find an expression of $u$ that maximizes $\tilde{\mathcal{E}}$. $w$ is the relative

price of ranking. The optimum point for $u$ satisfies first order condition, i.e., $\nabla\tilde{\mathcal{E}}(u) = 0$, this gives us:

$$\nabla\tilde{\mathcal{E}}(u) = \frac{\frac{\mathcal{C}_1 + \frac{1}{\nu(u)}}{\mathcal{C}_2(t) + \frac{1}{\nu(u)}} \frac{\frac{1}{\nu(u)^2} e^u (\mathcal{C}_2(t) - \mathcal{C}_1)}{(\mathcal{C}_1 + \frac{1}{\nu(u)})^2}(u + w) - \log\frac{\mathcal{C}_2(t) + \frac{1}{\nu(u)}}{\mathcal{C}_1 + \frac{1}{\nu(u)}}}{(u + w)^2}.$$

By first order condition,

$$\nabla\tilde{\mathcal{E}}(u^\star) = 0 \Rightarrow \frac{(\mathcal{C}_1 - \mathcal{C}_2(t))e^{u^\star}}{(\mathcal{C}_1 + \frac{1}{\nu(u^\star)})(\mathcal{C}_2(t) + \frac{1}{\nu(u^\star)})\nu(u^\star)^2}(u^\star + w) = \log\frac{\mathcal{C}_1 + \frac{1}{\nu(u^\star)}}{\mathcal{C}_2(t) + \frac{1}{\nu(u^\star)}}$$

$$\iff \frac{e^{u^\star}}{e^{u^\star} - 1}\left(\frac{1}{\mathcal{C}_2(t)\nu(u^\star) + 1} - \frac{1}{\mathcal{C}_1\nu(u^\star) + 1}\right)(u^\star + w) = \log(\mathcal{C}_1\nu(u^\star) + 1) - \log(\mathcal{C}_2(t)\nu(u^\star) + 1).$$

Define

$$g(\nu(u), \mathcal{C}) := \log(\mathcal{C}\nu(u) + 1) + \frac{\nu(u) + 1}{\nu(u)}\frac{1}{\mathcal{C}\nu(u) + 1}(\log(\nu(u) + 1) + w).$$

Consider an approximation of $g(\nu(u), \mathcal{C})$ which is valid for large $t$,

$$\tilde{g}(\nu, \mathcal{C}) := \log(\mathcal{C}\nu) + \frac{1}{\mathcal{C}}\frac{1}{\nu}(\log\nu + w).$$

Since $\tilde{g}(\nu(u^\star), \mathcal{C}_1) = \tilde{g}(\nu(u^\star), \mathcal{C}_2(t))$, for small $\mathcal{C}_2(t)$, $\nu(u^\star) \approx \frac{w}{\mathcal{C}_2\log\frac{\mathcal{C}_1}{\mathcal{C}_2}}$ which results in $u^\star \approx \log\frac{w}{\mathcal{C}_2(t)\log\frac{\mathcal{C}_1}{\mathcal{C}_2(t)}}$. ∎