[Reviews · NeurIPS 2019]

Reviewer 1



The paper considers the problem of mixed linear regression: in this problem, an algorithm is given access to n data samples (x_i, y_i) with possibly corrupted labels, where each y_i is one of the m possible linear functions of x_i, i.e., y_i = x_i^T theta_j for j in {1,...,m} (but the algorithm does not know which one). The goal of the algorithm is to determine vectors theta_1,..., theta_m. A straightforward but computationally inefficient (the complexity is exponential in d) approach to solving this problem is by using Least Trimmed Squares (LTS), which tries to identify the best fit vector (in terms of least squares) over all possible subsets of the data points of a particular, predefined size. To address this issue, the paper proposes using an alternative, simple, algorithm called Iterative Least Trimmed Squares (ILTS), which is similar to other algorithms that have been used for related problems in the literature, as acknowledged in the paper. The algorithm is essentially alternating minimization: it alternates between (1) finding the best set of a given size tau * n, given the least squares solution from the previous iteration and (2) solving least squares over the set determined in (1). This is a natural and very easy to implement algorithm, and I can see it being used if its convergence properties and ways in which to set the parameters are better understood. The main result of the paper provides sufficient conditions (in terms of the number of samples n, number of ground truth vectors m, a notion of separation of the data) for linear convergence of ILTS, under adversarial corruptions of the labels. These results are mainly meaningful in the setting where the feature vectors come from an isotropic Gaussian distribution and can be generalized to subgaussian distributions. This result requires that the algorithm is initialized close to a ground truth vector, which makes this convergence result local. The paper then provides a global version of the algorithm for isotropic Gaussian feature vectors. If I understood this part correctly, this result leverages robust PCA to find a subspace spanned by the ground truth vectors, constructs an eps-net over this subspace, and then applies ILTS to each point from the subspace. Overall, I find the topic of the paper to be interesting, and there are some neat ideas in the analysis, but I have a few questions/concern that would need be addressed before I could consider increasing the score: -- How important is it to know tau? How would one go about setting this parameter in practice? How does it relate to the fraction of corruptions? -- Looking at Table 1, compared to [14], the main difference is in the trade-off between sample and computational complexity -- for global convergence results, one pays more in the computation to have a lower sample complexity. [Is this trade-off inherent?] I think this should be explicitly discussed in the paper, for a fair comparison to related work. -- I believe that the claims about nearly-optimal computation need to be toned down, since the computation is really high for global guarantees. Stating that this is nearly optimal would need to be supported with the appropriate lower bounds for the trade-off between sample-efficiency and computation. -- Computation of the epsilon-net to get global guarantees seems like an overkill, which makes me doubt the usefulness of the main result (ILTS and its local analysis). -- I had a very hard time following Section 3.1 (Preliminaries) and in particular Definition 3 and the text surrounding it. Please consider revising this part. -- There are too many different constants in the statements of technical results in Section 4, which makes it very hard to understand the settings in which the results are useful. I suggest (at the very least) including some examples. While the problem is well-motivated, clearly introduced, and most of the introduction is well-written, I found the technical sections to be hard to follow. There is a lot of notation, that is often inconsistent (e.g., * in the superscript is used to define quantities corresponding to both corrupted and uncorrupted samples). I had a hard time understanding some of the assumptions (e.g., about the separation between the feature vectors). I also feel the language can be improved. Other specific and minor comments: -- Throughout: computation (complexity/lower bound) -> computational -- Line 61: tau -> tau *n -- Table 1: Compare -> Comparison -- Sigma_(j) in Table 1 is not defined at this point -- What is the meaning of the dash in Table 1 for the number of samples in the global case? -- Please add adequate references to the literature in Lines 81-82 -- Line 108: "of any previous work study" -- Line 109: "fine analysis" -> fine-grained analysis? -- Line 151: it is not clear at this point what it means for an algorithm to succeed -- Line 155: use either "for all" or \forall -- using both is superfluous -- Please use different patterns/markers in Figure 1 -- the figure is hard to read on a printout and would also probably be hard to read for colorblind people -- Definition 2: sigma_max and sigma_min are not defined -- Line 167-168: what does it mean that the prediction error is large due to the X? -- c_j is introduced in Theorem 7 but not used

Reviewer 2



Update: I have read the authors' response and I am satisfied with the answers it provided. =========================================== This paper considers the problem of mixed linear regression, where training data comes from multiple ground truth signals. They establish nontrivial performance guarantees for the simple iterative trimmed least squares (ITLS) approach, which alternates between fitting a subset of the data and greedily re-choosing the subset of the data to fit on. First, they show that under certain moderately strong assumption, this procedure convergence linearly to one of the ground truth signals. Second, they show that by starting this procedure with exponentially (in the number of of ground truth signals) many initializations, the entire model can be recovered. This apparently improves upon and extends the best known guarantees for this setting. I find the paper generally well-written and convincing; I believe it would make a nice contribution to NeurIPS. However, I am not sufficiently familiar with the related work to confidently estimate the significance of the results. I also did not carefully check the (technically involved) proofs in the supplementary material. Hence my low confidence score. Below are some additional comments and questions. 1) The main result is rather difficult to sparse and requires reading a number of technically dense pages. I believe that additional informal statements of Theorems 7 and 11 will significantly improve readability. 2) What is c_0 in Theorem 7? I could not find its definition. 3) Theorem 7 does not say anything about the probability 1-delta with which the results holds and how it factors into the complexity (does it affect n? \kappa?). I would appreciate explicit dependence on delta in the theorem statement. A similar comment holds for Lemma 5. Alternatively, you may fix delta to be poly(1/d) (the large parameter in the problem), but this would probably add an additional factor of log d to the expression for n. 4) The notation tau_{(j)} in Definition 3 is confusing, as by the previous notation tau_{(j)} should mean the fraction of examples belonging to signal j (similarly to S_{(j)} vs. S_{(j)}^\star), but you use is it to denote something else. Please use another notation. 5) What do you mean by "centered sphere" in line 5 of Algorithm 2? 6) The "Mixed linear regression" paragraph in page 3 features some overly long and difficult to parse sentences, for example in lines 105--108. 6) Some typos: - Line 144: size subset -> "" - Line 157: literatures -> literature - Line 173: "with size \tau_{(j)}^\star" -> "with size n \tau_{(j)}^\star" - Line 204: Affine -> The affine

Reviewer 3



I have read the author response. I still believe the lack of numerical experiments leave a lot of room for more insights to be generated, but I recommend publication even as-is. This is a well-written paper, which provides a theoretical analysis of an important algorithm under misspecification. Lack of numerical experiments (see below) is my main complaint. Given space constraints, I recommend publication at Neurips as is, although a thorough numerical evaluation of the algorithms---see suggestions below---would make the paper more impactful. Authors provide intuition for theoretical results, which I found to be quite helpful. Below are a few suggestions, comments, and questions to further improve the paper. General comments: 1) In Theorem 7, is there a sense of optimality for \gamma^*? Is there a sense of a minimal recoverable corruption level in the mixed linear regression setting? If the authors have a conjectured sense of this bound, a comparison with that given by the theorem would be helpful. 2) A numerical evaluation of the proposed algorithms would strengthen the paper significantly. For both local and global recovery scenarios, a natural question is to see if trimmed least squares genuinely does not provide good performance. Given recent advances in integer programming, an analysis of how off-the-shelf solvers scale, and when ILTS truly wins would be helpful. Comments on exposition/writing: 1) Table 1: Explain what Q is, or forward reference to Definition 1. 2) The phrase "smallest" component is confusing since it refers to the component weights. Being more specific (in general) would help with clarity of the exposition. 3) Some intuition for Lemma 6 would be helpful, and I believe it can be presented after Definition 3 to allow better understanding. 4) If space allows, a brief description of the gradient descent result should be included in the main text.

[Author Response · NeurIPS 2019]

We thank all the reviewers for the helpful comments. Here, we address the main concerns raised by the reviewers.

**To Reviewer # 4**

- [Additional discussion related to $\tau$] Any $\tau$ that is less than the ground truth one is sufficient; in parctice, one can start with a large $\tau$ and decay its value until the algorithm finds a parameter that fits $\tau n$ samples. That being said, of course a better estimate of $\tau$ means ILTS needs fewer samples. Additionally, the fraction of corruptions should always be no larger than a constant fraction of the interested component. Otherwise, it is impossible to identify if a mixture component comes from an adversary or not.

- [Compare with [14]] The computational complexity of [14] is **not** cheaper than ours. More specifically, in the "computation" column, the $n$ in each row represents the sample complexity of the corresponding algorithm. For [14], the $n$ in term $nd$ has an exponential dependency on $m$. We will make this clearer in the revised version.

- [Claim about nearly-optimal computation] We would like to point out that the computation used in the global step is not due to any trade-off between sample-efficiency and computation. Instead, it is related to the hardness of the problem in the general setting. Also, from a practical standpoint, our algorithm is easily parallelizable (while [14] is not).

- [About $\epsilon$-net] The $\epsilon$-net is applied to a $O(m)$-dimensional subspace, where $m$ is the number of mixtures. The mixed linear regression problem gets much harder as $m$ gets larger. In this general setting, $\epsilon$-net may not be an overkill. One intuition (also mentioned in [14]) is that recovering $m$ Gaussian mixture models in general requires at least exponential in $m$ number of samples. We consider the general setting, and hope the local/global analysis in this paper can shed light on understanding the structure of the problem. That being said, if additional assumptions (e.g., all components are orthogonal to each other) are added to the problem, there may exist more efficient approaches.

- [Writing of technical part and notations] We will keep improving the presentation of the technical part according to the reviewer's suggestion. In the current version, we try to make our notation self-contained: for example, the $\star$ in the superscript is used to denote "oracle knowledge" $- \theta_{(j)}^\star$ is the ground truth parameter for the $j$-th mixture component, $S_{(j)}^\star$ is the ground truth index set of samples that belong to the $j$-th mixture, $R^\star$ is the underlying index set of adversary samples.

**To Reviewer # 5**

- [Additional informal statements of Theorems 7 and 11 to improve readability.] We thank the reviewer for the kind suggestion. We will give an informal statement of our main results in the introduction section to improve readability.

- [$c_0$ in Theorem 7 ] $c_0$ is a constant such that $\kappa_t < 1$, and such a $c_0$ corresponds to an upper bound on $c_j$, i.e., the local region. We will add more detailed description to explain this Theorem in our revised version.

- [Explicit dependency on failure probability $\delta$] High probability in all our arguments means with probability at least $1 - n^{-c}$ (i.e., $\delta = n^{-c}$) for any given constant $c$. We did not write out the dependency explicitly because it contributes an additive factor of $\log d$ and is not the dominant term in the final expression for $n$. More intuitively speaking, our sample complexity is in the form of $d + \log(1/\delta)$, instead of $d \log(1/\delta)$. That being said, adding the dependency on $\delta$ would make our results more sensible, and we will add it in our revised version.

- [Typos] We will fix the typos and correct the confusing sentences as pointed out by the reviewer. "centered sphere" in line 5 of Algorithm 2 means the sphere with origin as its center.

**To Reviewer # 6**

- [Optimality of $\gamma^\star$] Our main convergence theorem holds when the corruption ratio is a constant fraction of the size of the interested mixture component. This corruption level is order-wise optimal. That being said, it is interesting to study other algorithms that can tolerate larger corruption ratio than ILTS does, in the mixed linear regression setting.

- [Numerical evaluations] We agree with the reviewer that adding rigorous numerical evaluations for ILTS and compare it with others in terms of scalability can be very helpful. We would like to add this section in our future versions. For the current paper, we would like to focus on the theoretical results, which may provide insights to understanding the mixed linear regression problem and the ILTS algorithm.

[Meta-Review · NeurIPS 2019]

This paper studies mixed linear regression and give a number of results. Under various deterministic conditions, they show that given a sufficiently warm start, iterative trimmed least squares converges to the true directions quickly. Their algorithm continues to work in the presence of adversarial corruptions. However the warm start is required to be quite close to the true solution. They give an SVD based initialization procedure that works in the non-noisy setting and when the examples come from a gaussian distribution. There has been much recent progress on mixed linear regression. Perhaps the most closely related paper is the work of Li-Liang. The main point of debate among the reviewers was, in light of what is known, how surprising are the results? Still, the results fill in some gaps in what was known for an important problem.